# *Bacillus thuringiensis* Vip1 Functions as a Receptor of Vip2 Toxin for Binary Insecticidal Activity against *Holotrichia parallela*

**DOI:** 10.3390/toxins11080440

**Published:** 2019-07-25

**Authors:** Jianxun Geng, Jian Jiang, Changlong Shu, Zeyu Wang, Fuping Song, Lili Geng, Jiangyan Duan, Jie Zhang

**Affiliations:** 1School of Life Sciences, Shanxi Normal University, Linfen 041004, China; 2State Key Laboratory for Biology of Plant Diseases and Insect Pests, Institute of Plant Protection, Chinese Academy of Agricultural Sciences, Beijing 100193, China

**Keywords:** *Bacillus thuringiensis*, Vip1Ad and Vip2Ag binary toxin, *Holotrichia parallela*, binding

## Abstract

*Bacillus thuringiensis* is a well-known entomopathogenic bacterium that produces vegetative insecticidal proteins (Vips, including Vip1, Vip2, Vip3, and Vip4) during the vegetative phase. Here, we purified Vip1 and Vip2 from *B. thuringiensis* and characterized the insecticidal effects of these protoxins. Bioassay results showed that a 1:1 mixture of Vip1Ad and Vip2Ag, purified by ion-affinity chromatography independently, exhibited insecticidal activity against *Holotrichia parallela* larvae, with a 50% lethal concentration value of 2.33 μg/g soil. The brush border membrane (BBM) in the midgut of *H. parallela* larvae was destroyed after feeding the Vip1Ad and Vip2Ag mixture. Vacuolization of the cytoplasm and slight destruction of BBM were detected with Vip2Ag alone, but not with Vip1Ad alone. Notably, Vip1Ad bound to BBM vesicles (BBMVs) strongly, whereas Vip2Ag showed weak binding; however, binding of Vip2Ag to BBMV was increased when Vip1Ad was added. Ligand blotting showed that Vip2Ag did not bind to Vip1Ad but bound to Vip1Ad-t (Vip1Ad was activated by trypsin), suggesting the activation of Vip1Ad was important for their binary toxicity. Thus, our findings suggested that Vip1Ad may facilitate the binding of Vip2Ag to BBMVs, providing a basis for studies of the insecticidal mechanisms of Vip1Ad and Vip2Ag.

## 1. Introduction

White grubs are important insect pests that severely damage the roots of many crops, including soybeans, corn, peanuts, turf, and some vegetables [1]. Furthermore, *Holotrichia parallela* larvae are not easy to control owing to their soil-dwelling habit. Currently, the management of *H. parallela* larvae is highly dependent on the use of chemical pesticides [2], and many efforts are being made to develop environment friendly means of controlling pests. In China, *H. parallela* has infected large areas of peanut, soybean, and sweet potato crops, causing significant reductions in crop yields and great economic losses [3].

As an alternative to chemical pesticides, *Bacillus thuringiensis* (Bt) biopesticides have been employed in pest control for decades [4], and many *H. parallela* specific Bt strains are currently being evaluated [5,6,7]. Different insecticidal proteins were produced by *B. thuringiensis*, such as the vegetative insecticidal proteins (Vip) during the vegetative phase and the crystal proteins (Cry) during the sporulation phase of growth [8]. Cry protein consist of three-domain Cry (3d-Cry) family, the Mtx-like Cry family, and Bin-like Cry family. Among these, the 3d-Cry family represents the biggest family, which are globular molecules containing three distinct domains connected by single linkers [8]. Cry proteins receive their mnemonic names and four hierarchical ranks depending on their primary sequence identity. The first rank is a different number that is given if the protein shares less than 45% identity with all the other Cry proteins. The second rank is a capital letter that is given if the protein shares less than 78% but more than 45% identity with other Cry proteins. The third rank gives a lowercase letter to distinguish proteins that share more than 78% identity and less than 95% with other Cry proteins [8]. Previous data have shown that three domain toxins, i.e., Cry8E, Cry8F, Cry8G and Cry8-like from Bt strains, are toxic to *H. parallela* larvae [6,7,9]. So far, 15 Vip1 proteins, 20 Vip2 proteins, 111 Vip3 proteins, and one Vip4 protein (the nomenclature used for Cry toxin is also applicable to Vip toxin), have been reported in the following website (http://www.lifesci.sussex.ac.uk/home/Neil_Crickmore/Bt/). Recently, two Vip toxins, Vip1Ad and Vip2Ag (abbreviations for Vip1Ad1 or Vip2Ag1), were found in Bt strain HBF-18 (CGMCC 2070). Vip1Ad and Vip2Ag exhibited binary toxicity against *H. parallela* larvae [7]. However, the insecticidal mechanisms of Vip1Ad and Vip2Ag against *H. parallela* larvae have not been elucidated yet.

Currently, there are two different type activity models for binary toxins. In the first model (the “A-B” model), components A and B, for example, *Cholera* and *Escherichia coli* toxins, form a complex before binding to the cell surface in solution [10]. In the second model (the “A + B” model), components A and B do not form aggregates before binding to the cell surface [11]; instead, subunit A binds to the membrane, and the membrane-bound subunit A then provides a pathway for subunit B to enter the cytoplasm of the target cell [12,13,14]. In previous studies, an assumed activity model of Vip1 and Vip2 was proposed, suggesting that Vip1 was activated by a trypsin-like protease in the midgut [15]. The monomer of Vip1 formed oligomers, and the oligomers could recognize specific receptors in the midgut. Thus, Vip1 provided a pathway for Vip2 to enter the cytoplasm. The Vip2 domain was able to catalyze the transfer of the ADP-ribose group from NAD to actin, prevent its polymerization, and thus inhibit microfilament network formation [16,17]. However, there was not sufficient data to support this model. 

Accordingly, in this study, we investigated the interactions among Vip1Ad, Vip2Ag, and brush border membrane vesicles (BBMVs). The results of our study are expected to provide insights into the insecticidal mechanisms of Vip1/Vip2.

## 2. Results

### 2.1. Preparation of Vip1Ad and Vip2Ag

The recombinant strains HDVIP1 and HDVIP2 were grown in LB medium and cultural supernatants were analyzed by sodium dodecyl sulfate polyacrylamide gel electrophoresis (SDS-PAGE). Clear bands were observed at the corresponding molecular weights (Vip1Ad [90 kDa], Vip2Ag [40 kDa]; Appendix A). The Vip1Ad and Vip2Ag were purified by ion exchange chromatography method. The results for Vip1Ad showed two obvious elution peaks (Figure 1A). SDS-PAGE results showed that Vip1Ad was well enriched and purified in the first elution peak (Figure 1B). In contrast, Vip2Ag showed only one elution peak (Figure 1C), and SDS-PAGE results showed that Vip2Ag was well enriched in the elution peak (Figure 1D). 

### 2.2. Bioassay of Vip1Ad and Vip2Ag against H. parallela Larvae

The insecticidal bioassay by using purified Vip1Ad and Vip2Ag proteins showed that a mixture of Vip1Ad and Vip2Ag (molar ratio 1:1) exhibited insecticidal activity against *H. parallela* larvae, with 50% lethal concentration (LC_50_) values of 2.33 μg/g soil (Table 1). However, the corrected mortality of *H. parallela* larvae, individually, are 36.67% and 26.67% when the concentration of Vip1Ad and Vip2Ag is 50 μg/g soil suggesting that the LC_50_ should be higher than 50 μg/g. The corrected mortality is calculated by the mortality of Vip1Ad/Vip2Ag subtracted the mortality of negative control (larvae treated with Tris-HCl containing diet).

### 2.3. Histopathological Effects of Vip1Ad/Vip2Ag on H. parallela Larvae

The results of histopathological analyses by using transmission electron microscope (TEM) showed slender microvilli in an ordered arrangement when larvae had ingested phosphate-buffered saline (PBS; a negative control; Figure 2A). The midgut showed significant changes in the microvillus, cytoplasmic vacuolation after the larvae had ingested HBF-18 (CGMCC 2070, a positive control; Figure 2D). Compared with the negative control, destruction of the midgut manifested as vacuolization of the cytoplasm and abscission of microvilli when *H. parallela* larvae fed a mixture of Vip1Ad and Vip2Ag (30 μg/g soil; molar ratio 1:1; Figure 2E). However, no obvious destruction of the BBM was observed in the midgut of *H. parallela* larvae fed Vip1Ad toxin alone (Figure 2B). Vacuolization of the cytoplasm and slight destruction of BBM were detected after *H. parallela* larvae fed Vip2Ag alone (Figure 2C), as compared with the negative control.

### 2.4. Analysis of Binding of Vip1Ad and Vip2Ag to BBMVs of H. parallela

In order to verify the binding of Vip1Ad and Vip2Ag with BBMVs of *H. parallela* larvae, the BBMVs were extracted and the aminopeptidase N (APN) enzyme activity of BBMVs was 6.35 times the midgut homogenate, confirming its applicability in subsequent experiments. Aminopeptidase N (APN) activity was monitored because it is a marker for the BBMV and because APN is a putative receptor for many Bt toxins [18]. The Vip1Ad can be activated by trypsin into Vip1Ad-t that contains the Vip1Ad activated toxin and oligomers showing heat and SDS resistance (Panel A in Figure 3), but Vip2Ag cannot be activated by trypsin (as showed in Appendix A). Binding assay between Vip1Ad or Vip1Ad-t and BBMVs was performed as followed, showing that both Vip1Ad and Vip1Ad-t bound to BBMVs. Interestingly, Vip1Ad formed an activated and oligomeric mode which was similar to Vip1Ad-t, indicating that the Vip1Ad could be triggered into oligomers when incubation with BBMVs by activation of protease embedded in BBMVs (Panel B in Figure 3). The interaction between Vip1Ad and BBMVs was confirmed by saturation binding assay through ELISA. The binding of Vip1Ad and BBMVs can be saturated, suggesting that the binding is specific and the affinity constant (*Kd*) was calculated to 17.86 ± 4.38 nM (Figure 3, Panel C). In following part, the binding assay of Vip2Ag and BBMVs was performed, showing that Vip2Ad has a weak binding with BBMVs but the binding signal was increased significantly when Vip1Ad was added (Figure 3, Panel D). The interesting result suggested that Vip2Ag binds to BBMVs under the assistance of Vip1Ad and Vip1Ad could function as a receptor of Vip2Ag on BBMVs probably. 

### 2.5. Analysis of the Interaction between Vip1Ad and Vip2Ag

In order to study the interaction between Vip1Ad and Vip2Ag, we analyzed the binding between Vip1Ad and Vip2Ag toxins by ligand blotting and dot blotting assay. Vip1Ad-t-monomer was purified by Superdex-75 gel filtration chromatography from Vip1Ad-t (Appendix A). Firstly, Vip2Ag (1 μg) was loaded onto SDS-PAGE and transferred to PVDF membranes which incubated with Vip1Ad, Vip1Ad-t and Vip1Ad-t-monomer (50 nM), separately. The interaction was found between Vip2Ag and Vip1Ad-t but not with Vip1Ad, suggesting that the activation of Vip1Ad is necessary for binding (Figure 4A). Meanwhile, the Vip1Ad-t-monomer was purified from Vip1Ad-t, which contained the monomer of the Vip1Ad activated toxin only, while, the Vip1Ad-t-monomer showed no binding with Vip2Ag. The results were verified by a dot blot. Two microliters of Vip1Ad, Vip1Ad-t and Vip1Ad-t-monomer were loaded evenly onto nitrocellulose (NC) membranes directly and incubated with 50 nM Vip2Ag showing that Vip2Ag interacted with Vip1Ad-t but no binding was found between Vip1Ad and Vip1Ad-t-monomer with Vip2Ag (lower panel in Figure 4B). Adversely, two microliters of Vip2Ag were loaded evenly onto nitrocellulose (NC) membranes and incubated with Vip1Ad, Vip1Ad-t and Vip1Ad-t-monomer showing that Vip1Ad-t interacted with Vip2Ag but no binding was found between Vip1Ad, and Vip1Ad-t-monomer with Vip2Ag (upper panel in Figure 4B). The binding between Vip1Ad-t and Vip2Ag were found but no binding was detected with Vip1Ad or Vip1Ad-t monomer and Vip2Ag, which indicated that Vip2Ag exerts its function by interaction with the oligomers of Vip1Ad-t.

However, no binding was found when the ligand-blot was performed between Vip1Ad and Vip2Ag. Vip1Ad-t or Vip1Ad-t-monomer (1 μg) were analyzed by SDS-PAGE, transferred to PVDF membranes and incubated with Vip2Ag (50 nM). The results indicated that the structure of Vip1Ad-t is important for the interaction with Vip2Ag, assuming that the binding could be fulfilled by the structural oligomers of Vip1Ad-t (Figure 4C). All results above demonstrated that trypsin activation is necessary for the binding of Vip1Ad-t with Vip2Ag and the binding is performed by oligomers of Vip1Ad-t probably. 

### 2.6. Ultracentrifugation Analysis of Oligomerization of Vip1Ad-t and Vip2Ag

The SDS-PAGE results showed that Vip1Ad-t included monomers and oligomers. Thus, we performed additional analyses of the states of oligomerization using ultracentrifugation. The continuous size-distribution and sedimentation coefficient distribution showed a distinct sharp peak corresponding to the monomer and a relatively broad distribution of larger species corresponding to oligomers (Figure 5A). The oligomers were composed of trimers (red arrow), tetramers (yellow arrow), pentamers (green arrow), hexamers (blue arrow), and heptamers (purple arrow). Accordingly, different oligomerization states were formed by digestion of Vip1Ad. The binding between Vip1Ad-t and Vip2Ag was evaluated using ultracentrifugation. The c(s) sedimentation coefficient distribution showed a distinct sharp peak corresponding to the monomer and a relatively broad distribution of larger species formed (Figure 5B).

## 3. Discussion

*Holotrichia parallela* is an important agricultural and forestry pest that is difficult to control. Cry8 toxins exhibit activity against *H. parallela* [5,6,19]. Previous studies have demonstrated that Vip1Ad and Vip2Ag toxins show excellent activity against *H. parallela* larvae [7]. Thus, further analyses of the mechanisms of action of Vip1 and Vip2 against *H. parallela* are essential for the application of these protoxins.

High concentrations of Vip1Ad and Vip2Ag expressed from Bt were found in this study, reaching to 51.20 and 57.45 g/L, respectively. Secretory proteins and high expression gave Vip1Ad and Vip2Ag outstanding characters for investigation and pest control. The Cry1Ac protein containing HD-73 strain is one of the prominent strains of Bt insecticide at present. We compared the expression of Vip1Ad and Vip2Ag with Cry1Ac protein. The SDS-PAGE result showed that the protein expression of Vip1Ad, Vip2Ag was higher than that of Cry1Ac under same amount of strain culture products (Appendix A). The results of optical density analysis showed that the expression of Vip1Ad was 1.36 times of that of Cry1Ac and the expression of Vip2Ag was 1.60 times of that of Cry1Ac.

A complex of Vip1 and Vip2 toxins exhibited high insecticidal activity against Coleoptera, and Hemiptera [7,20], although the mechanisms mediating these effects of Vip1 and Vip2 have not previously been evaluated. In this study, we found that Vip1Ad binds to BBMVs through an activation version including activated toxin and oligomers (Figure 3B). The binding is specific because it can be saturated in ELISA and *Kd* is 17.86 ± 4.38 nM (Figure 3C). This result was similar to that of Vip1Ae, which has been shown to bind to BBMVs of cotton aphids, but without being digested by these BBMVs [20]. The reason for this could be related to the presence of trypsin-like protease embedded in BBMVs extracts. Other reports have not provided evidence of digestion of Vip1 protoxin by BBMVs.

There are two types of binary toxin, i.e., “A-B” and “A + B”. Bioinformatics analyses have shown that Vip1Ad and Vip2Ag belong to the actin-ADP-ribosylating binary toxin family, as similar as “A + B” type binary toxins, for which A and B components do not interact in solution [21,22,23]. Each polypeptides of binary toxin have a unique function. One subunit binds to the membrane, and the membrane-binding A subunit provides a pathway for the B subunit into the cell [10]. A and B subunits are required for maximal activity against insects. In our study, we found that Vip1Ad acted as an A subunit binding to BBMVs of *H. parallela*; although Vip2Ag as the B subunit could also bind to BBMVs of *H. parallela*, the degree of binding was low. When Vip1Ad was added, the binding of Vip2Ag increased. Similar results were observed in histopathological analysis. Vip2Ag alone led to little damage to BBMVs of *H. parallela* (Figure 2C), but the harmful effects were obviously increased when Vip1Ad was added. Destruction of the midgut manifested as vacuolization of the cytoplasm and abscission of microvilli. These effects were similar to the vacuolization previously described in *Helicoverpa armigera* [24,25], and *Prays oleae* [26] larvae treated with Cry toxins and in *Spodoptera littoralis* [27] fed Vip3A protoxin. This demonstrated that binary toxins Vip1Ad and Vip2Ag, Cry, and Vip3A toxins had similar histopathological effects in the control of agricultural pests.

Previous studies have reported that Vip1Ac could be activated by trypsin and the activated Vip1Ac formed oligomers containing seven monomers [15]. In our study, Vip1Ad could be activated by trypsin also and the activation could trigger oligomerization (Figure 3A). The activation form of 01Ad, Vip1Ad-t, interacted with Vip2Ag (Figure 4A,B). Otherwise, the binding of Vip1Ad-t with Vip2Ag cannot be detected when Vip1Ad-t was loaded on the gel for SDS-PAGE and following the ligand blot (Figure 4C) suggesting that the natural structure is important for binding of Vip1Ad-t with Vip2Ag. 

Our findings were verified by the ultracentrifugation analysis of oligomerization of Vip1Ad-t and Vip2Ag (Figure 5). The oligomers of Vip1Ad-t were found in consist of hexamers mainly and also other oligomerization states, such as heptamers (Figure 5A). Otherwise, the oligomers with larger size were found when analyzing oligomerization of Vip1Ad-t together with Vip2Ag (Figure 5B). Oligomers of 605 kDa might be two Vip2Ag binding to one heptamer of Vip1Ad-t, and oligomerization at 720 kDa may be five Vip2Ag binding one heptamer of Vip1Ad-t. Of course, the larger oligomers may also have other binding forms. Accordingly, Vip1 may function as a receptor for Vip2, resulting in binary toxin insecticidal activity against *H. parallela*. Our findings suggested that the interaction of Vip1Ad-t with Vip2Ag could be fulfilled by the oligomers of Vip1Ad-t. However, direct evidence is lacking because no binding between oligomers of Vip1Ad-t and Vip2Ag was detected, so the assumption needs to be confirmed further. There are two hypotheses after binding; Vip1 could help Vip2 pass through the membrane by conformational change and function inside cells; or, Vip1 helps Vip2 access the BBM and some receptors act with Vip2 and exert its function on the membrane. Obviously, further studies are needed to determine the roles of this interaction in the mechanism of action of binary toxins. 

Our results clarify the interactions among Vip1, Vip2, and BBMVs of *H. parallela*. Our findings provided a framework for understanding the insecticidal mechanism of Vip1/Vip2. The results of this study highlighted that Vip2Ag led to little damage to BBM of *H. parallela* and that the degree of damage increased dramatically when Vip1Ad was added. Thus, Vip1Ad may be used as a receptor to help Vip2Ag bind to BBMVs. Additional investigations are required to fully elucidate the insecticidal mechanisms of Vip1Ad and Vip2Ag.

## 4. Materials and Methods 

### 4.1. Expression of Vip1Ad and Vip2Ag 

Recombinant strains HDVIP1 and HDVIP2 (the genes of *vip1Ad* and *vip2Ag* were cloned and expressed under the control of the *cry3Aa* promoter of pSTK in the acrystalliferous Bt strain HD 73^−^, respectively) of *B. thuringiensis* were selected on LB agar plates containing 50 μg/mL kanamycin and incubated at 30 °C for 12 h [7]. A single colony was transferred to a test tube containing 5 mL LB medium (50 μg/mL kanamycin), and the tube was incubated at 30 °C with shaking at 220 rpm for 12 h. Four milliliters of bacterial solution were then transferred to a triangular flask containing 400 mL LB medium (50 μg/mL kanamycin), and this flask was incubated at 30 °C with shaking at 220 rpm for 24 h. The fermentation broth was centrifuged (9000× *g* for 15 min at 4 ℃), and the supernatant was collected. The concentrations of Vip1Ad and Vip2Ag protein extracts were determined by ImageJ Software using different concentrations of bovine serum albumin (BSA) as a standard. The standard curves were drawn with different concentrations of BSA standard protein. Then Vip1Ad and Vip2Ag were quantified by way of the optical density method.

### 4.2. Purification of Vip1Ad and Vip2Ag 

Vip1Ad and Vip2Ag were purified by the ion exchange chromatography method [28]. All chromatography purification runs were performed on ÄKTA-25 system (GE Healthcare Life Sciences, Pittsburgh, PA, USA) at 25 °C and the column effluent was monitored by absorbance at 280 nm. After dilution 10-fold using buffer A (20 mM Tris-HCl, pH 8.0), HiTrap Q HP and HiTrap SP HP columns were used for the purification of Vip1Ad and Vip2Ag, respectively. The sample was loaded onto the column that was equilibrated with buffer A and eluted with a linear gradient against buffer B (20 mM TrisCl pH 8.0, 1 M NaCl). The gradient extended from 0% to 100% of the buffer B concentration in 20 min. [29]. SDS-PAGE analysis was performed after purification of Vip1Ad and Vip2Ag by ion exchange chromatography. The protein concentration was measured with a Pierce BCA Protein Assay kit (Thermo Scientific, Waltham, MA, USA), using bovine serum albumin (BSA) as the standard.

### 4.3. Bioassay

*Holotrichia parallela larvae* of 5-day-old, obtained from Cangzhou Academy of Agricultural and Forestry Science, Hebei, were used in the bioassay. The bioassay diet for *H. parallela* larvae was prepared as previously described [7]. The concentrations (5 and 50 μg/g soil) of Vip1Ad, Vip2Ag and the combination of Vip1Ad and Vip2Ag were used to preliminary screening. Five appropriate concentrations (0.37, 1.11, 3.33, 10, 30 μg/g soil) of Vip1Ad, Vip2Ag and the combination of Vip1Ad and Vip2Ag were used to determine the LC_50_. Twelve larvae (three replicates) were fed with each concentration of toxin. A diet treated with Tris-HCl buffer was used as a negative control. The mortality of larvae was investigated after incubation for 14 days, and each assay was repeated at least twice.

### 4.4. Statistical Analysis

Statistical analyses were conducted using the SPSS software (version 13.0, SPSS Inc., Chicago, IL, USA). Analysis of variance (ANOVA) was performed using a general linear model and mean values of LC_50_ were compared using Fisher’s least, with its associated 95% confidence interval. *P* < 0.05 represents a significant difference.

### 4.5. Preparation and Sectioning of Insect Tissues for Histopathology Observation

Third instar larvae of *H. parallela* were starved overnight before being exposed to Vip1Ad (30 μg/g soil) and Vip2Ag (30 μg/g soil), and PBS (1 mM KH_2_PO_4_, 10 mM Na_2_HPO_4_, 137 mM NaCl, 2.7 mM KCl [pH 7.4]) was used as a negative control. *Bacillus thuringiensis* strain HBF-18(CGMCC 2070), which has been shown to be toxic to this insect, was used as a positive control [6]. The toxins were included in the diet, as described previously [5]. After 3 days of exposure, the midguts were dissected on ice and placed in 2.5% glutaraldehyde to solidify for 24 h at room temperature. After washing for 2 h in PBS, the tissues were transferred to 1% osmic acid, allowed to solidify for an additional 2 h, and then washed for 2 h in distilled water. The tissues were dehydrated in increasing ethanol concentrations, infiltrated in 100% acetone, and embedded in resin. Five-micrometer-thick sections were obtained, placed on round copper nets coated with uranyl acetate for 15 min, and washed in distilled water. The sections were then incubated with 0.02 M NaOH and immediately transferred to lead citrate. After washing in distilled water, the copper nets with tissue sections were stored at room temperature until analysis. Images were captured by transmission electron microscopy (H-7500; Hitachi Limited, Hitachi, Japan) [30].

### 4.6. Preparation of BBMVs of H. parallela and Purity of BBMV Determination

The larvae were reared with germinated wheat in soil until the third instar. The midgut tissue was dissected. BBMVs were prepared by the magnesium precipitation method [31]. The purity of BBMVs was determined by estimating the enrichment of APN enzyme specific activity compared with that in the initial midgut tissue homogenate [18].

### 4.7. Binding of Vip1Ad and Vip2Ag to BBMVs of H. parallela by Western Blotting

One micrograms Vip1Ad (or Vip2Ag) were incubated with twenty micrograms BBMVs of *H. parallela* for 1 h at room temperature in 100 μL binding buffer (0.1% BSA in PBST (PBS containing 0.1% Tween-20). The Vip1Ad and Vip2Ag used for this assay were centrifuged 10 min at 14,000× *g* before incubation. Binding reactions were stopped by centrifugation for 10 min at 14,000× *g*, after which the BBMV pellet containing the bound toxin was washed twice with 0.1 mL ice-cold binding buffer [32]. Pellets were solubilized in sample buffer (10 μL), heat denatured for 10 min at 100 °C, and loaded onto SDS-PAGE gels. After electrophoresis, the separated proteins were transferred to PVDF membranes using a Genescript Protein Transfer System (Genescript, Nanjing, China). The PVDF membranes were incubated with anti-Vip1Ad antibodies (anti-Vip2Ag antibodies; 1:5000 dilution) and rabbit anti-mouse IgG conjugated to horseradish peroxidase (HRP) as secondary antibodies (1:20,000 dilution) [33]. Before blocking, the membranes were washed with 0.1% Tween-20 in PBS solution and developed with Super Signal chemiluminescence substrate (Thermo).

### 4.8. Protein Biotinylation

Vip1Ad protein was labeled with EZ-Link Sulfo-NHS-SS-Biotin (Thermo Fisher Scientific, Rockford, IL, USA) as per the manufacturer’s instructions in phosphate-buffered saline (PBS) buffer (137 mM NaCl, 2.7 mM KCl, 10 mM Na_2_HPO_4_, 1.8 mM KH_2_PO_4_, pH 7.4) at room temperature for 2 h. Free biotin was removed by using a desalting column (GE Healthcare) with 20 mM Tris-HCl, pH 8.0. Purified biotinylated Vip1Ad proteins concentration was measured with a Pierce BCA Protein Assay kit (Thermo Scientific, USA).

### 4.9. Binding of Vip1Ad and Vip2Ag to BBMVs of H. parallela by ELISA

One milligram BBMVs of *H. parallela* in PBS were immobilized on 96-well plate for 12 h at 4 °C, followed by five washes with PBST. The plate was blocked with PBST (PBS containing 0.1% Tween-20) with 2% BSA for 2 h at 37 °C. The plate was incubated with biotinylated Vip1Ad (0, 5, 10, 20, 40, 80, and 160 nM) for 1 h at 37 °C. After washing with PBST for five times, the biotinylated Vip1Ad were detected with streptavidin-HRP (1:20,000 dilution) for 1 h at 37 °C. The HRP enzymatic activity was revealed with 40 mg of *o*-phenylenediamine and 18 mL of H_2_O_2_ in 100 mL of 100 mM NaH_2_PO_4_, pH 5.0. The enzymatic reaction was stopped with 1 M HCl, and the absorbance was read at 490 nm with an LKB Ultraspec II spectrophotometer (Amersham Biosciences, Munich, Germany). Binding data were analyzed and plotted with SigmaPlot v.13.0 software (Systat Software, San Jose, CA, USA).

### 4.10. Ligand Blot Analysis of the Interactions of Vip1Ad, Vip1Ad-t and Vip2Ag

Vip1Ad (10 μg) was incubated with trypsin (10:1 w:w ratio) at 37 °C for 2 h. The product was named Vip1Ad-t. Vip1Ad, Vip1Ad-t and Vip1Ad-t-monomer (20 μL) with sample loading buffer (5 μL) were subjected to SDS-PAGE on 10% gels. After electrophoresis, the separated protein was transferred to PVDF membranes. Subsequent to protein transferring, the PVDF membranes were blocked with PBS containing 2% Tween-20 for 1 h at room temperature [34]. Next, the membranes were incubated with 50 nM Vip2Ag, probed with anti-Vip2Ag antibodies (1:5000 dilution), and incubated with rabbit anti-mouse IgG conjugated to HRP (1:20,000 dilution). In the same way, we transferred Vip2Ag to the PVDF membrane and incubated it with Vip1Ad or Vip1A-t. Finally, it was probed by anti-Vip1Ad antibodies (1:5000 dilution), and incubated with rabbit anti-mouse IgG conjugated to HRP (1:20,000 dilution). Before blocking, the membranes were washed with 0.1% Tween-20 in PBS and developed with Super Signal chemiluminescence substrate [35].

### 4.11. Dot Blot Analysis of the Binding of Vip1Ad, Vip1Ad-t to Vip2Ag

Two microliters of Vip1Ad and Vip1Ad-t were pipetted evenly onto two strips of NC membranes. The strips were incubated at room temperature until the droplets on the bands dried. The NC strips were then blocked by incubating in PBS containing 2% Tween-20 for 1 h at room temperature. Next, the strips were incubated with 50 nM Vip2Ag, probed with anti-Vip2Ag antibodies (1:5000 dilution), and incubated with rabbit anti-mouse IgG conjugated to HRP (1:20,000 dilution). In the same way, two microliters of Vip2Ag were pipetted evenly onto two strips of NC membranes, respectively. The NC strips were then blocked by incubating in PBS containing 2% Tween-20 for 1 h at room temperature. Next, the strips were incubated with 50 nM Vip1Ad or Vip1Ad-t. The anti-Vip2Ag antibodies (1:5000 dilution) were incubated. The signal was detected by rabbit anti-mouse IgG conjugated to HRP (1:20,000 dilution). Before chemiluminescence, the membranes were washed with 0.1% Tween-20 in PBS and developed with Super Signal chemiluminescence substrate [36].

### 4.12. Analysis of Oligomerization of Vip1Ad by Ultracentrifugation

The analysis platform for ultracentrifugation was a Proteome Lab XL-1 solution protein interaction analysis system provided by the Institute of Biophysics, Chinese Academy of Sciences. Samples of Vip1Ad-t were condensed to 2.5 mg/mL. The buffer was 20 mM Tris-HCl (pH 8.0). The purity of the samples was determined based on the standard of analysis for ultracentrifugation. Analysis of the sedimentation of protein samples was performed under interferometric light at a speed of 40,000× *g* at 20 °C set for ultracentrifugation [37].

## Figures and Tables

**Figure 1 toxins-11-00440-f001:**
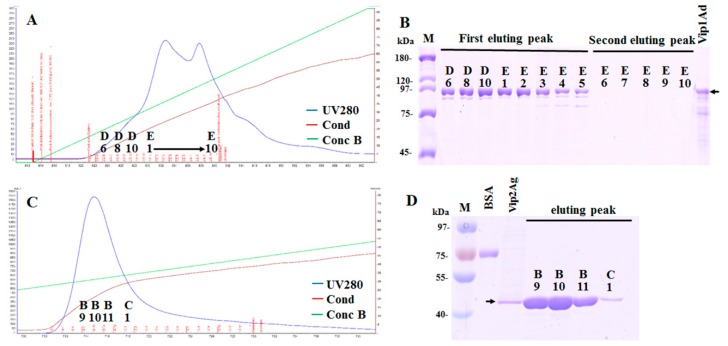
Analysis of vegetative insecticidal proteins (Vip)1Ad and Vip2Ag purified by ion-exchange chromatography. (**A**) Step gradient elution profile of Vip1Ad purified using a HiTraq Q HP column. (**B**) Sodium dodecyl sulfate polyacrylamide gel electrophores (SDS-PAGE) analysis of purification of Vip1Ad using an anionic column. (**C**) Step gradient elution profile of Vip2Ag purified using a HiTraq SP HP column. (**D**) SDS-PAGE analysis of the purification of Vip2Ag using a cationic column. In Figure 1A, C, the black arrows indicate the HDVIP1 and HDVIP2 supernatants, blue line: UV280 nm, red line: conductivity, green line: percentage of eluant.

**Figure 2 toxins-11-00440-f002:**
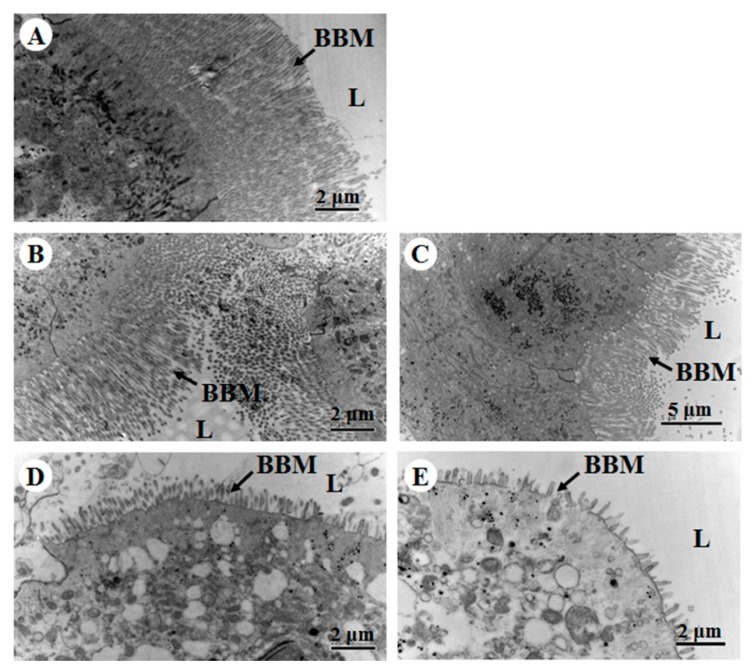
Histopathological effects of Vip1Ad and Vip2Ag binary toxin on the midgut of *H. parallela* larvae observed by transmission electron microscope (TEM). Sections of the midgut epithelium from larvae fed (**A**) phosphate-buffered saline (PBS, negative control), (**B**) Vip1Ad (30 μg/g soil), (**C**) Vip2Ag (30 μg/g soil), (**D**) Bt strain HBF-18 (positive control), or (**E**) a mixture of Vip1Ad and Vip2Ag (molar ratio 1:1; 30 μg/g soil). L, lumen; BBM, brush border membrane.

**Figure 3 toxins-11-00440-f003:**
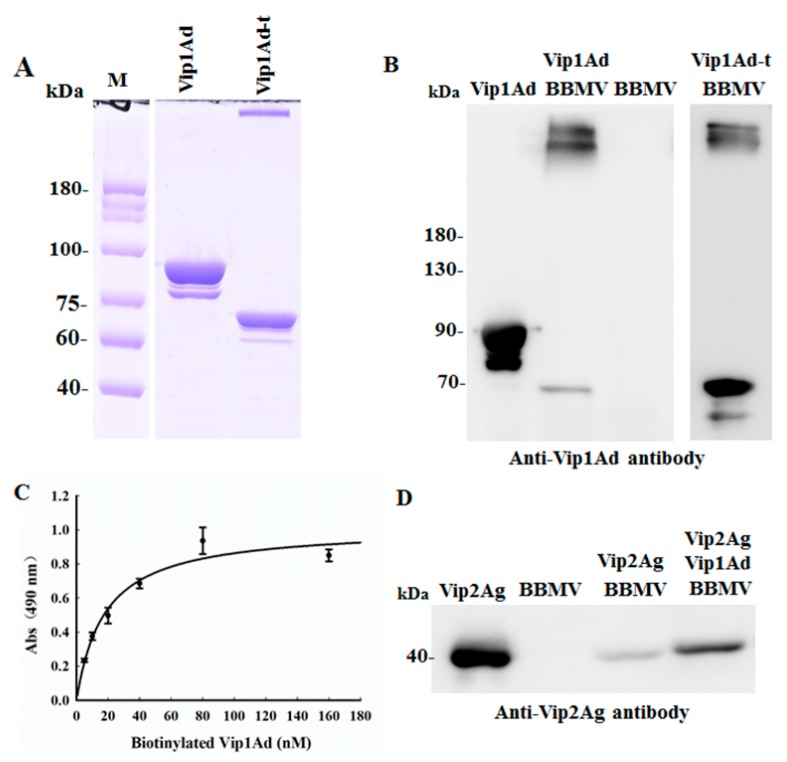
Binding assays for Vip1Ad, Vip2Ag and BBM vesicles (BBMVs) of *H. parallela*. (**A**) SDS-PAGE analysis of Vip1Ad activation by trypsin. (**B**) Western blot analysis of binding between Vip1Ad, Vip1Ad-t and BBMVs. The Vip1Ad on gel as a positive control and BBMV on gel as a negative control. (**C**) ELISA analysis of binding between Vip1Ad and BBMVs. (**D**) Western blot analysis of binding between Vip2Ag and BBMVs. Vip2Ag on gel as positive control and BBMV as negative control.

**Figure 4 toxins-11-00440-f004:**
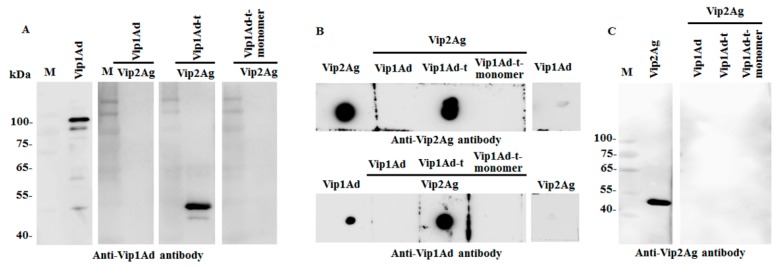
Analysis of the interaction between Vip1Ad and Vip2Ag. (**A**) Ligand blot analysis of the binding of Vip1Ad, Vip1Ad-t and Vip1Ad-t-monomer with Vip2Ag. The Vip1Ad on gel as a positive control. (**B**) Panel above: Dot blot analysis of the binding of Vip2Ag with Vip1Ad, Vip1Ad-t and Vip1Ad-t-monomer. The Vip2Ag on membrane as a positive control and Vip1Ad as a negative control. Panel lower: Dot blot analysis of the binding of Vip1Ad, Vip1Ad-t and Vip1Ad-t-monomer with Vip2Ag. The Vip1Ad on membrane as a positive control and Vip2Ag as a negative control. (**C**) Ligand blot analysis of the Vip2Ag with Vip1Ad, Vip1Ad-t-monomer or Vip1Ad-t. Vip2Ag on gel as a positive control. Vip1Ad and Vip2Ag are protoxins, Vip1Ad-t is the products of activated Vip1Ad by trypsin, Vip1Ad-t-monomer is monomer fragment (70 kDa) purified from Vip1Ad-t.

**Figure 5 toxins-11-00440-f005:**
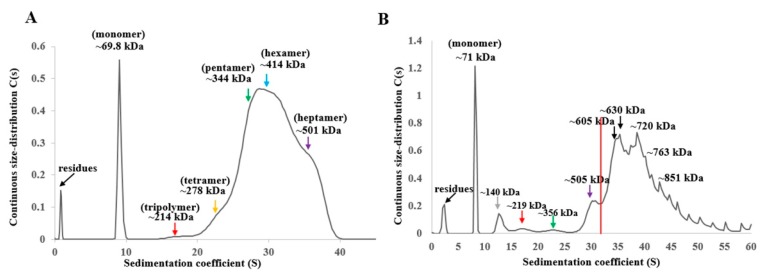
Ultracentrifugation analysis of the oligomerization of Vip1Ad-t and Vip2Ag. (**A**) Ultracentrifugation analysis of the oligomerization of Vip1Ad. (**B**) Ultracentrifugation analysis of the binding between Vip2Ag and Vip1Ad-t. Different oligomers were shown by various colors. Grey arrows indicate deployer, red arrows indicate trimers, yellow arrows indicate tetramers, green arrows indicate pentamers, blue arrows indicate hexamers and purple arrows indicate heptamers.

**Table 1 toxins-11-00440-t001:** Bioassays of insecticidal activity of Vip1Ad and Vip2Ag toxins against *H. parallela* larvae.

Protoxin	LC_50_ (μg/g soil) ^a^	Slope ± SEM ^b^
Vip1Ad	>50	
Vip2Ag	>50	
Vip1Ad + Vip2Ag	2.33 (1.14–4.08)	1.08 ± 0.21

^a^ The confidence limits at 95% are given in parentheses. ^b^ Mean ± standard error of the mean.

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
