# Peer review of "Bacillus thuringiensis* Vip1 Functions as a Receptor of Vip2 Toxin for Binary Insecticidal Activity against *Holotrichia parallela"

_toxins, 2019, doi:10.3390/toxins11080440_

Round 1

Reviewer 1 Report

The authors have addressed my concerns. 

Author Response

Thank you very much for your comments. 

Reviewer 2 Report

The authors present a study investigating the interaction between the Bacillus thuringiensis toxins Vip1Ad and Vip2Ag. They demonstrate fairly convincingly that Vip1Ad is needed for Vip2Ag to exert its toxic effect on Holotrichia paralella, and thus act as a A+B -type toxin pair. My major concern is that the paper abounds with mis-labelings, missing information and incorrect figures, which leads to a very confusing read (see below for specific instances). In addition, though the English of the manuscript is largely acceptable, there are points where the English is poor and words appear to be missing (for example, line 10 in the abstract). The article must undergo more rigorous proofreading before it is in a publishable form. In addition I have the following major comments:

The introduction seem too brief. The authors do not properly introduce the Cry toxins and the Vip toxins, nor do they explain the interplay between these toxin types. Further, they do not explain the classification of the Vip toxins - what are Vip1Ad and Vip2Ag, and how do they relate to other toxins of these classes? Currently this is all left for the reader to find out elsewhere and makes for a confusing start.

In section 2.5, the authors show the interaction of Vip1Ad and Vip2Ag with BBMVs by western blot. They conclude that oligomeric VipA1d interacts with BBMVs. This is supported by the presented data. Interestingly (and no mention is made of this in the paper), the oligomers appear to be highly stable as they can be boiled in SDS for 10 minutes. The authors conclude that " binding of monomer was very weak". Based on the presented data, I am not sure the monomer binds at all. The band seen for the "monomer" is approximately 70 kDa, smaller than the monomer of purified Vip1Ad in the previous lane. This is also smaller than the trypsin-activated Vip1Ad-T (~80 kDa). At least three scenarios could explain this band: i) it is a digestion product of VipAd produced by a protease in the BBMVs that falls of an oligomer but is not involved in binding, or ii) it is the active form of Vip1Ad (analogous to Vip1Ad-T) that binds to BBMVs as a complex, but due to extensive heating in SDS, some of the complexes have dissociated (but the complex is still needed for binding), or iii) it is indeed a Vip1Ad-T-like monomer that is genuinely able to bind to BBMVs. This band could be characterised by mass spectrometry and compared to Vip1Ad-T. Furthermore, this oligomer bands seen in the blot should also be subjected to mass specterometric analysis to see if they are the same species as the ~70 kDa band, which would distinguish between options i and ii. To see whether monomers do actually bind to BBMVs, the authors could use their purified Vip1Ad-T monomers in a similar binding experiment.

Figure 5 is a mess. The left-hand panel in Fig. 5A is labled "VipAd monomer" - but as far as I can see, Vip1Ad  is always monomeric (see Figs 1, 2, 3 and Fig. 5A). Surely they mean "Vip1Ad-T monomer"? This same problem affects panels B and C as well. In Fig. 5B, as I understand it, Vip2Ag was run in SDS-PAGE and transferred to the membrane, then overlaid with Vip1A species? This looks like a single membrane - how were the three different proteins incubated with the one membrane? If it is a splice of three different membranes, the splicing needs to be made obvious to show the manipulation (as it is for the control blot of Vip1Ad). The same holds for panel C. In addition, control blots without the overlaid proteins are needed to show that the Vip2Ag antibody does not bind to Vip1Ad-T without Vip2Ag. Similar controls should be added for the dot blots.

Supplementary Figs 1 and 2 are not what is stated in the main text. There it reads, "Figure S1: Analysis of Vip1Ad and Vip2Ag digested by trypsin" and "Figure S2: Analysis 324 of the monomer of Vip1Ad was purified by gel filtration chromatography". But the actual supplementary figures show "Figure S1. Ligand blot analysis of the Vip2Ag binding to Vip1Ad, Vip1A-T and Vip1Ad monomer" and "Figure S2. Analysis of Vip2Ag protoxin digested by trypsin." The former does not show any binding to Vip1Ad-T, and there is no discussion about this surprising finding in the text.

Minor comments (excluding typos etc)

line 10: Are the yields of the proteins necessary for the abstract?

line 64: the concentrations given here seem overly precise to me.

line 69: here it is claimed that the concentrations were estimated by ImageJ, but in the Materials & Methods (line 251) it says the BCA method was used

section 2.2: I do not think this is necessary. Most of the information is also in the Materials & Methods, and the purification image could be in supplementary. It would be sufficient to say that the proteins were purified by IEX at the beginning of the next section.

Lines 97 and 108-109: nowhere here does it say that the images are electron micrographs. What is the size of the scale bar?

Line 113: A little more context is needed here; this is not explained until Materials & Methods

Line 129: again, a little more context would be useful

Lines 160, 163 and 173: why use the term "tripolymers", when "trimers" is more usual?

Line 167: what is the evidence for homo-/hetero-oligomer formation in Fig. 6B? Or is this just assumed?

Line 184: As mentioned in Major Comment 2, the association of monomers with BBMVs has not been fully proven.

Line 259: How was the LD50 value calculated?

Sections 4.8 and 4.9: the experiments were performed both ways (i.e. immobilising either Vip1Ad species or Vip2Ag) - this should be mentioned here.

Author Response

Response to Reviewer 2 Comments

Point 1: The introduction seems too brief. The authors do not properly introduce the Cry toxins and the Vip toxins, nor do they explain the interplay between these toxin types. Further, they do not explain the classification of the Vip toxins - what are Vip1Ad and Vip2Ag, and how do they relate to other toxins of these classes? Currently this is all left for the reader to find out elsewhere and makes for a confusing start.

Response 1: Thanks for your suggestion. We added one paragraph for introduction of production of Bt toxins, which could be easier for understanding by readers.

Point 2: In section 2.5, the authors show the interaction of Vip1Ad and Vip2Ag with BBMVs by western blot. They conclude that oligomeric VipA1d interacts with BBMVs. This is supported by the presented data. Interestingly (and no mention is made of this in the paper), the oligomers appear to be highly stable as they can be boiled in SDS for 10 minutes.

Response 2: Thanks for your suggestion. Yes, Vip1Ad may interacts with BBMVs firstly and function as the recptor for Vip2Ag through its oligomeric structure, but it’s a indirect conclusion saying that it’s oligomers functioned. The assumption came from the results that Vip1Ad-t bound to BBMVs instead of Vip1Ad and the heterologous found in ultracentrifugation analysis with both toxins.

In general, the oligomers of Cry toxins are not stable higher that 70, but the oligomers of Vip1Ad is indeed more stable than Cry toxins from our results.  

Point 3: The authors conclude that " binding of monomer was very weak". Based on the presented data, I am not sure the monomer binds at all. The band seen for the "monomer" is approximately 70 kDa, smaller than the monomer of purified Vip1Ad in the previous lane. This is also smaller than the trypsin-activated Vip1Ad-T (~80 kDa). At least three scenarios could explain this band: i) it is a digestion product of VipAd produced by a protease in the BBMVs that falls of an oligomer but is not involved in binding, or ii) it is the active form of Vip1Ad (analogous to Vip1Ad-T) that binds to BBMVs as a complex, but due to extensive heating in SDS, some of the complexes have dissociated (but the complex is still needed for binding), or iii) it is indeed a Vip1Ad-T-like monomer that is genuinely able to bind to BBMVs. This band could be characterised by mass spectrometry and compared to Vip1Ad-T. Furthermore, this oligomer bands seen in the blot should also be subjected to mass specterometric analysis to see if they are the same species as the ~70 kDa band, which would distinguish between options i and ii. To see whether monomers do actually bind to BBMVs, the authors could use their purified Vip1Ad-T monomers in a similar binding experiment.

Response 3: Thanks for your suggestion. After careful consideration, we decided not to show the results of Vip1Ad monomer in Fig 3 because it’s easy to make readers confused with this term and also because the monomer is not that rigorous that the monomer could form oligomers when interaction with BBMVs so it’s difficult to idetify monomer when incubated with BBMV independently.

We aggree with you about the first scenario that Vip1Ad could be activated when incubated with BBMVs because there are proteases embedded in BBMVs extracts, suggesting that Vip1Ad incubated with BBMVs suffered same process with activition by trypsin. That’s why the Vip1Ad-t and Vip1Ad have same bands when incubation with BBMVs.

Point 4: Figure 5 is a mess. The left-hand panel in Fig. 5A is labled "VipAd monomer" - but as far as I can see, Vip1Ad is always monomeric (see Figs 1, 2, 3 and Fig. 5A). Surely, they mean "Vip1Ad-T monomer"? This same problem affects panels B and C as well. In Fig. 5B, as I understand it, Vip2Ag was run in SDS-PAGE and transferred to the membrane, then overlaid with Vip1A species? This looks like a single membrane - how were the three different proteins incubated with the one membrane? If it is a splice of three different membranes, the splicing needs to be made obvious to show the manipulation (as it is for the control blot of Vip1Ad). The same holds for panel C. In addition, control blots without the overlaid proteins are needed to show that the Vip2Ag antibody does not bind to Vip1Ad-T without Vip2Ag. Similar controls should be added for the dot blots.

Response 4: Thanks for your suggestion. According to your suggestion, we have modified the figure 5. The term of “Vip1Ad-t monomer” was used for new revision. For ligand-blotting assay, the membranes are separated actually and we have adjusted the distance among membranes. Meanwhile the negative control of Vip1Ad or Vip2Ag for detection with antibody were added to prove that there are cross-reaction between toxins and antibodies.

Point 5: Supplementary Figs 1 and 2 are not what is stated in the main text. There it reads, "Figure S1: Analysis of Vip1Ad and Vip2Ag digested by trypsin" and "Figure S2: Analysis 324 of the monomer of Vip1Ad was purified by gel filtration chromatography". But the actual supplementary figures show "Figure S1. Ligand blot analysis of the Vip2Ag binding to Vip1Ad, Vip1A-T and Vip1Ad monomer" and "Figure S2. Analysis of Vip2Ag protoxin digested by trypsin." The former does not show any binding to Vip1Ad-T, and there is no discussion about this surprising finding in the text.

Response 5: Thanks for your suggestion. We have adjusted the supplementary materials according to the comments of the review, and we have also explained your doubts in the discussion section.

Point 6: line 10: Are the yields of the proteins necessary for the abstract?

Response 6: Thanks for your suggestion. Yes The abstract is not the right place to tlak about this in detail and we have made modification.

Point 7: line 64: the concentrations given here seem overly precise to me.

Response 7: Thanks for your suggestion, probably it’s the unit make you confused but the result it’s no problem which calculated by the ImageJ with BSA standard.

Point 8: line 69: here it is claimed that the concentrations were estimated by ImageJ, but in the Materials & Methods (line 251) it says the BCA method was used.

Response 8: Thanks for your suggestionWe have added to Method 4.1 how to determine the protein concentration with BSA and ImageJ.

Point 9: Section 2.2: I do not think this is necessary. Most of the information is also in the Materials & Methods, and the purification image could be in supplementary. It would be sufficient to say that the proteins were purified by IEX at the beginning of the next section.

Response 9: Thanks for your suggestion, but we would like to present this result in results because the Vip1 and Vip2 proteins are secreted into medium directly and the quantity of expression are very high, also the big amount purified proteins could be achieved with easy process We believed it’s a very interesting strain for invetigation. We have added this part to the 2.1 according to the review comments.

Point 10: Lines 97 and 108-109: nowhere here does it say that the images are electron micrographs. What is the size of the scale bar?

Response 10: We added the information about TEM here as well as the scale bar.

Point 11: Line 113: A little more context is needed here; this is not explained until Materials & Methods

Response 11: Thanks for your suggestion. Here we add the purpose of doing the experiment to facilitate readers to understand.

Point 12: Line 129: again, a little more context would be useful.

Response 12: Thanks for your suggestion. In order to study the interaction between Vip1Ad and Vip2Ag, we analyzed the binding between Vip1Ad and Vip2Ag toxins by ligand blotting and dot blotting assay. We have added this part into 2.5.

Point 13: Lines 160, 163 and 173: why use the term "tripolymers", when "trimers" is more usual?

Response 13: Thanks for your suggestion. We have changed the tripolymers to trimers in this article.

Point 14: what is the evidence for homo-/hetero-oligomer formation in Fig. 6B? Or is this just assumed?

Response 14: Thanks for your suggestion. It’s a hypothesis obviously. We’ve moved the assumption to the discussion.

Point 15: Line 184: As mentioned in Major Comment 2, the association of monomers with BBMVs has not been fully proven.

Response 15: Thanks for your suggestion. We also realized this problem, so we added that experiment in figure 3-B.

Point 16: Line 259: How was the LC50 value calculated?

Response 16: Thanks for your suggestion. First, we counted the mortality of each protein concentration insect, then statistical analyses were conducted using the SPSS 13.0 software. The detailed content was added to 4.3

Point 17: Sections 4.8 and 4.9: the experiments were performed both ways (i.e. immobilising either Vip1Ad species or Vip2Ag) - this should be mentioned here

Response 17: Thanks for your suggestion. We added another experimental method to sections 4.9 and 4.10.

Reviewer 3 Report

General comments:

The aim of the study is to further investigate the mechanism of action of Vip1Ad and Vip2Ag proteins. Their data suggest that both proteins are able to interact with BBMV of H. parallela, though they do not show whether the binding was specific or not, as they do not perform competition experiments. Furthermore, they showed that the totla binding of the Vip2Ag increased when Vip1Ad was added to the reaction. Finally, the also showed that both proteins were able to interact and that the interaction is happening with the oligomeric structure of the Vip1Ad.

In general, the language of manuscript needs a minor spell check. However, the results are not clear explained, and part of the results are presented in the discussion section rather than in the results section. Thus, this make more difficult to follow the main findings of the manuscript. Some major issues must to be addressed by the authors:

Sections 2.1 and 2.2

This two sections can be combined, as they explain how the authors expressed and purified both samples. It would be great to clarify that both Vip proteins were purified as a protoxins (section 2.2).

Section 2.4

It could be great to add the concentration used of each toxin in the text. Why the authors used this high concentration to check the histopathological effects?

Why some histopathological effects could be observed when larvae were treated with Vip2Ag but not with Vip1Ad?

Section 2.5

Lines 113-115: The measurement of APN activity can be written in the material and methods section rather that in the results section (as the authors said “confirming its applicability in subsequent experiments”).

It would be interesting to clarify that the protoxin was used in the binding experiments and that after binding two fragments were obtained, one that could be an oligomeric structure (>180 kDa) and the other band which corresponds to the activated fragment.

The authors observed binding but as they did not perform any competition experiment how can we be sure that the binding is specific and that it is not an artifact (e.g precipitation during the assay)?

Section 2.6

It would be great if the authors explain their hypothesis to clarify why they perform the subsequent experiments. (line 129-130).

The figure S1 should be included in the main manuscript. In general, this section is difficult to follow as the authors did not explain why these experiments have been performed or why they used two different methodologies to address the same issue. Only in the discussion section (lines 208-212) you can understand why the authors perform the dot blot. They claim that the discrepancies found between the SDS-PAGE and Western and the Dot blot can be due because the structure of the oligomer can be modifying when the Vip1Ad oligomer is loaded in a SDS-PAGE. For that reason, I consider that the Figure S1 should be in the main text.

The term monomer sometimes is confusing; I will say the trypsin-activated form of 70 kDa. Please clarify the nomenclature used (Vip3Ad-T, monomer…).

Section 2.7

Line 156: Figure S2-A is missing in the supplementary material. In contrast, the Fig. S2 B it is included in the supplementary material but it is not cited in the text.

Line 160 (and along the text): Why the authors used the term trypolymers instead of tetramers?

Why all these variants were not visualized in the gel? Is the high molecular weight band (oligomer in the gel) in the resolving part of the gel or in the stacking part?

Material and methods section:

Section 4.6

The results of the APN activity can be added in this section as I have suggested above.

Section 4.7 and 4.8

Please indicate the amount of Vip proteins used for the assays (lines 288 and 301). Line 300 “Vip1Ad in Trsi-HCl (pH 8.0)” which is the concentration of the buffer? The methodology has to be described with detail in all the sections.

Why the authors do not add the reciprocal experiment? I mean they do not say that the same experiment (section 4.8 and 4.9) were performed loading/ adding Vip2Ag and then probing with Vip1Ad.

Minor comments:

Lines 40-41: The sentence is difficult to understand. Why the authors add the info in the brackets?

Lines 59-61: Part of the text is basically material and methods and should not be included in the results section.

Lines 64-65: The amount of protein obtained is not relevant, so it is not necessary to include in the main text.

Lines 67-70: The authors write “The concentration of Vip1Ad and ….as a standard” from my point of view this info it is not relevant in the figure. The main aim is to show that in the expression system used they get both Vip proteins. I will not include this info.

Lines 72-75: Again, the authors wrote material and methods. Please removed this info.

Line 101: “negative control, Destruction” (D should be in lowercase)

Lines 90-91: “However, the corrected mortality of …” what it is the meaning of corrected mortality? Please rewrite the sentence to clarify it.

Line 129; “deeply, The Vip1Ad…” The T should be lowercase.

Line 131: “we” (in capital letter).

Line 164: “was the evaluated using …” remove “the”

Line 165: Rewrite the sentence as “The sedimentation coefficient (S) …”

Line 221 and 223: Change cytomembrane for membrane.

Line 222: “or, Vip1 helps Vip2 access to the BBMVs and some…” Do you mean BBM (brush border membrane)?

Line 227: results of this study highlighted that Vip2Ag led to little damage to BBMVs of H. parallela and that…” do you mean damage to the gut not to BBMVs don’t you?

Line 254 and 271: At the beginning of a sentence you have to write the full name of the species (H. parallela_ Holotrichia parallela and B. thuringiensis_ Bacillus thuringiensis).

Author Response

Response to Reviewer 3 Comments

Point 1: Sections 2.1 and 2.2. These two sections can be combined, as they explain how the authors expressed and purified both samples. It would be great to clarify that both Vip proteins were purified as a protoxins (section 2.2).

Response 1: Thanks for your suggestion. We’ve combined 2.1 and 2.2 ino one section and added the purification process for both protoxins.

Point 2: Section 2.4. It could be great to add the concentration used of each toxin in the text. Why the authors used this high concentration to check the histopathological effects?

Response 2: Thanks for your question. We used 30 μg/g soil becaise it’s ralatively higher than the LC50 of Vip1 and Vip2 (50 μg/g soil). To keep same, the concentration of combination of Vip1 and Vip2 was settled at 30 μg/g soil.

Point 3: Why some histopathological effects could be observed when larvae were treated with Vip2Ag but not with Vip1Ad?

Response 3: Thanks for your question. In previous studies of binary toxins and in our studies, Vip1Ad provides a pathway for Vip2Ag to enter cells, which itself does not kill cells. The main effect of killing cells is Vip2Ag, when the concentration of Vip2Ag is high, the chance of the Vip2Ag to contact with the intestinal wall is increased, this could be the reason why Vip2Ag mildly damaged the midgut cells.

Point 4: Section 2.5. Lines 113-115: The measurement of APN activity can be written in the material and methods section rather that in the results section (as the authors said “confirming its applicability in subsequent experiments”).

Response 4: Thanks for your suggestion. We have adjusted this part of the content into the material and methods section.

Point 5: It would be interesting to clarify that the protoxin was used in the binding experiments and that after binding two fragments were obtained, one that could be an oligomeric structure (>180 kDa) and the other band which corresponds to the activated fragment.

Response 5: Thanks for your suggestion. According to your suggestion, we have modified the figure 5 and clarified the two bands of Vip1Ad-t bingding with BBMVs

Point 6: The authors observed binding but as they did not perform any competition experiment how can we be sure that the binding is specific and that it is not an artifact (e.g precipitation during the assay)?

Response 6: Thanks for your suggestion. We couldn’t add competition assay because of the shortage of larvae and the limitaion of time for modification of the manuscript. However, we have some explanations for the specific binding: we added the saturation binding assay by ELISA we did before which could partially demonstrate the binding are specific. Also, Vip1Ad or Vip2Ag was centrifugated 10 min at 12000 ×g before binding assay everytime before binding assay. Therefore, the protein detected by the antibody is not precipitated by itself, but specifically binding.

Point 7: Section 2.6. It would be great if the authors explain their hypothesis to clarify why they perform the subsequent experiments. (line 129-130).

Response 7: Thanks for your suggestion. In order to study the interaction between Vip1Ad and Vip2Ag, we analyzed the binding between Vip1Ad and Vip2Ag toxins by ligand blotting and dot blotting assay. We have added this part into paragraph 2.5.

Point 8: The figure S1 should be included in the main manuscript. In general, this section is difficult to follow as the authors did not explain why these experiments have been performed or why they used two different methodologies to address the same issue. Only in the discussion section (lines 208-212) you can understand why the authors perform the dot blot. They claim that the discrepancies found between the SDS-PAGE and Western and the Dot blot can be due because the structure of the oligomer can be modifying when the Vip1Ad oligomer is loaded in a SDS-PAGE. For that reason, I consider that the Figure S1 should be in the main text.

Response 8: Thanks for your suggestion. We have adjusted the supplementary materials according to the comments of the review. Figure S1 have added in the main manuscript

Point 9: The term monomer sometimes is confusing; I will say the trypsin-activated form of 70 kDa. Please clarify the nomenclature used (Vip3Ad-T, monomer…).

Response 9: Thanks for your suggestion. Yes, the term of “Vip1Ad monomer” makes readers confused. We changed it to Vip1Ad-t monomer with explanation in manuscript. Vip1Ad is protoxin with a molecular weight of about 90 kDa. Vip1Ad-t is the product of trypsin activation of Vip1Ad protoxin including monomer (~70 kDa) and oligomers (>180 kDa). Vip1Ad-t monomer is the monomeric form of Vip1Ad activated toxin purified from Vip1Ad-t.

Point 10: Section 2.7. Line 156: Figure S2-A is missing in the supplementary material. In contrast, the Fig. S2 B it is included in the supplementary material but it is not cited in the text.

Response 10: I'm sorry for my mistake. This error was omitted from the last modification.

Point 11: Line 160 (and along the text): Why the authors used the term trypolymers instead of tetramers?

Response 11: Thanks for your suggestion. Thanks for your suggestion. We have changed the tripolymers to trimers in the revision.

Point 12: Why all these variants were not visualized in the gel? Is the high molecular weight band (oligomer in the gel) in the resolving part of the gel or in the stacking part?

Response 12: Thanks for your question. We sopposed that there are variants of oligomers on the SDS-PAGE gel with Vip1Ad, but they focus into one single band that cannot be separated on the gel.

Point 13: Section 4.6. The results of the APN activity can be added in this section as I have suggested above.

Response 13: Thanks for your suggestion. We have adjusted the APN activity into the material and methods section 4.6.  

Point 14: Section 4.7 and 4.8. Please indicate the amount of Vip proteins used for the assays (lines 288 and 301). Line 300 “Vip1Ad in Trsi-HCl (pH 8.0)” which is the concentration of the buffer? The methodology has to be described with detail in all the sections.

Response 14: Thanks for your suggestion. We have given a detailed description of the amount of protein in each test.

Point 15: Why the authors do not add the reciprocal experiment? I mean they do not say that the same experiment (section 4.8 and 4.9) were performed loading/ adding Vip2Ag and then probing with Vip1Ad.

Response 15: Thanks for your suggestion. We have added another experimental method to sections 4.9 and 4.10.

Point 16: Lines 40-41: The sentence is difficult to understand. Why the authors add the info in the brackets?

Response 16: I'm sorry for my mistake. The contents in parentheses have been deleted.

Point 17: Lines 59-61: Part of the text is basically material and methods and should not be included in the results section.

Response 17: Thanks for your suggestion. We have adjusted this part of the content into the material and methods.

Point 18: Lines 64-65: The amount of protein obtained is not relevant, so it is not necessary to include in the main text.

Response 18: Thanks for your suggestion, but we would like to present this result in results because the Vip1 and Vip2 proteins are secreted into medium directly and the quantity of expression are very high, also the big amount purified proteins could be achieved with easy process. We believed it’s a very interesting strain for further invetigation abour its promoter and secretion machnism.

Point 19: Lines 67-70: The authors write “The concentration of Vip1Ad and ….as a standard” from my point of view this info it is not relevant in the figure. The main aim is to show that in the expression system used they get both Vip proteins. I will not include this info.

Response 19: After considering your suggestions, we decided to move this figure of qutification to supplement figure because this result is not directly related to our conclusion.

Point 20: Lines 72-75: Again, the authors wrote material and methods. Please removed this info.

Response 20: Thanks for your suggestion. After consideration, we decided to put the protein expression and purification in 2.1, so this part of the content was retained in the materials and  methods.

Point 21: Line 101: “negative control, Destruction” (D should be in lowercase)

Response 21: I'm sorry for my mistake. We have rewrited the “D” to “d” in the manuscript.

Point 22: Line 129; “deeply, The Vip1Ad…” The T should be lowercase.

Response 22: I'm sorry for my mistake. We have rewrited the “T” to “t” in the manuscript.

Point 23: Line 131: “we” (in capital letter).

Response 23: I'm sorry for my mistake. We have rewrited the “w” to “W” in the manuscript.

Point 24: Line 164: “was the evaluated using …” remove “the”

Response 24: I'm sorry for my mistake. We have removed “the” it in the manuscript.

Point 25: Line 165: Rewrite the sentence as “The sedimentation coefficient (S) …”

Response 25: I'm sorry for my mistake. We have rewrited sentence with the sedimentation coefficient (S)  in the manuscript.

Point 26: Line 221 and 223: Change cytomembrane for membrane.

Response 26: I'm sorry for my mistake. We have replaced cytomembrane with membrane in the manuscript.

Point 27: Line 222: “or, Vip1 helps Vip2 access to the BBMVs and some…” Do you mean BBM (brush border membrane)?

Response 27: Yes, you are right, it’s BBM actually.Vip1 helps Vip2 access to the BBM and the BBM is brush border membrane.

Point 28: Line 227: results of this study highlighted that Vip2Ag led to little damage to BBMVs of H. parallela and that…” do you mean damage to the gut not to BBMVs don’t you?

Response 28: Thanks for your suggestion. Vip2Ag led to little damage to BBM not BBMVs. We’ve modified the term in the manuscript as you suggested.

Point 29: Line 254 and 271: At the beginning of a sentence you have to write the full name of the species (H. parallela_ Holotrichia parallela and B. thuringiensis_ Bacillus thuringiensis).

Response 29: I'm sorry for my mistake. We have modified similar errors in the full text.

Round 2

Reviewer 2 Report

The authors have submitted a revised version of their paper, "Bacillus thuringiensis Vip1 functions as a receptor of Vip2 toxin for binary insecticidal activity against Holotrichia parallela." The current version has addressed most of my earlier concerns; however, I still have some small comments.

The English language of the revised regions (marked in yellow) is substandard and needs to be proofread by a native-level English speaker.

Lines 48-49: The authors should make it explicit that the nomenclature used for Cry toxin subclassing also applies to Vip toxins.

Lines 126-27: "In order to verify the binding of Vip1Ad and Vip2Ag with BBMVs of H. parallela larvae, the BBMVs were extracted and the APN enzyme activity of BBMVs is 6.35 times as the midgut homogenate, confirming its applicability in subsequent experiments." Why is APN activity relevant? This is explained later in the Materials and Methods, but the reason for using this as a proxy should be here (as should the explanation of what APN is).

Line 129: I really think the authors should add "heat and SDS-resistant oligomers" to their description of Vip1Ad activated toxin. 

Lines 132-133: "Interestingly, Vip1Ad formed activation and oligomeric mode which is similar 132 to Vip1Ad-t, indicating that the Vip1Ad could be activated during incubation with BBMVs". By activation, do you mean proteolytic digestion? The only evidence for this is the weak band at ~70 kDa. To verify the the oligomers are also made from this "activated" form, you would need to check them by fully denaturing SDS-PAGE (see below). Something worth noting here or in the Discussion is that the oligomers of the BBMV-activated Vip1Ad are apparently more stable than those of Vip1Ad-t. One possible explanation for this is that there are several processing sites on Vip1Ad. A single processing leads to oligomer formation, but cleavage at another site(s) leads to to monomer (~70 kDa) formation (see comment 7). Trypsin may be more efficient at cleaving at the secondary site than the BBMV endogenous protease. Alternatively, the cleavage site may (trypsin vs. endogenous protease) could affect the stability of the oligomer. The authors could boil their samples in SDS + 6M urea (or even in the presence of formic acid) to fully denature the oligomers and see the size of the monomers. 

Lines 139-140: It might be interesting to check whether the presence of Vip2Ag has an effect on the activation/cleavage pattern of Vip1Ad. Again, I do not expect the authors to do this, this is just a suggestion. Note that there is a typo for Vip2Ag on line 140.

Line 353: Please provide the composition of PBST.

Line 354: How was biotinylation performed?

Author Response

Response to Reviewer 2 Comments

Point 1: The English language of the revised regions (marked in yellow) is substandard and needs to be proofread by a native-level English speaker.

Response 1: Thanks for your suggestion. The language has been improved which showed with red color.

Point 2: Lines 48-49: The authors should make it explicit that the nomenclature used for Cry toxin subclassing also applies to Vip toxins.

Response 2: Thanks for your suggestion. The modification has been made at Line 49-50 in red color.

Point 3: Lines 126-27: "In order to verify the binding of Vip1Ad and Vip2Ag with BBMVs of H. parallela larvae, the BBMVs were extracted and the APN enzyme activity of BBMVs is 6.35 times as the midgut homogenate, confirming its applicability in subsequent experiments." Why is APN activity relevant? This is explained later in the Materials and Methods, but the reason for using this as a proxy should be here (as should the explanation of what APN is).

Response 3: Thanks for your suggestion. The explanation was added in results section ar Line 126-128.

Point 4: Line 129: I really think the authors should add "heat and SDS-resistant oligomers" to their description of Vip1Ad activated toxin.

Response 4: Thanks for your suggestion. The explanation was added at Line 129.

Point 5: Lines 132-133: "Interestingly, Vip1Ad formed activation and oligomeric mode which is similar 132 to Vip1Ad-t, indicating that the Vip1Ad could be activated during incubation with BBMVs". By activation, do you mean proteolytic digestion? The only evidence for this is the weak band at ~70 kDa. To verify the the oligomers are also made from this "activated" form, you would need to check them by fully denaturing SDS-PAGE (see below). Something worth noting here or in the Discussion is that the oligomers of the BBMV-activated Vip1Ad are apparently more stable than those of Vip1Ad-t. One possible explanation for this is that there are several processing sites on Vip1Ad. A single processing leads to oligomer formation, but cleavage at another site(s) leads to to monomer (~70 kDa) formation (see comment 7). Trypsin may be more efficient at cleaving at the secondary site than the BBMV endogenous protease. Alternatively, the cleavage site may (trypsin vs. endogenous protease) could affect the stability of the oligomer. The authors could boil their samples in SDS + 6M urea (or even in the presence of formic acid) to fully denature the oligomers and see the size of the monomers.

Response 5: Thanks for your suggestion. Yes, the activation means proteolytic digestion.

I agree with you that it’s not sufficient to say the activated toxin and oligomers of Vip1Ad from incubation with BBMV are totally same with Vip1Ad-t from trypsin activated, but it’s also difficult to say that oligomers of the BBMV-activated Vip1Ad are apparently more stable than those of Vip1Ad-t because we repeat this assay for several times and it’s not always happen the same like we showed below. As we know, Western-blot is not a very good assay for qutification. Thus, for our conclusion in the manuscript, we concluded that the bands are similar between BBMV-activated Vip1Ad and Vip1Ad-t. We assumed that Vip1Ad could bind to BBMV by protealysis of protease on BBMVs according to many publishments saying that pore-forming toxins need to be activated .

Point 6: Lines 139-140: It might be interesting to check whether the presence of Vip2Ag has an effect on the activation/cleavage pattern of Vip1Ad. Again, I do not expect the authors to do this, this is just a suggestion. Note that there is a typo for Vip2Ag on line 140.

Response 6: Thanks for your suggestion. We did this experiment already as followed SDS-PAGE showed. The right panel showed activation of Vip1Ad and the right panel showed that the addition of Vip2Ag has no effect on the digestion of Vip1Ad.

Point 7: Line 353: Please provide the composition of PBST.

Response 7: Thanks for your suggestion. The composition of PBS is added in first appear at line 330, and PBST was showed in 350 at the position of forst appear in Matetrials and Method.

Point 8: Line 354: How was biotinylation performed?

Response 8: Thanks for your suggestion. We have added the method of biotinylation performed in Matetrials and Method at Line 361.

Reviewer 3 Report

In general, the authors have addressed all the issues that I made. However, as the review was done in a short time the modifications added in the main text can be improved (as an example the way of how they expressed the new info). It would be great if the authors can have improved the way the expressed the new sections included.

Some examples:

- Lines 75-76: Indicate that the yield obtained of each protein was high. This info “…and they are secretory proteins, which are easy to obtain” is part of the discussion not results. I totally agree on that it is important that the yield obtain was high but I think it would be great if you can put it in the discussion section and compare with other expression yields reported by other authors.

- Lines 90-91: “However, the corrected mortality of …” what it is the meaning of corrected mortality? Please rewrite the sentence to clarify it.

-Line 181: Please rewrite the sentence “... trans membraned to PVDF membranes…” Do you mean and transfer to PVDF membranes? In general, this section could be improved.

- Line 230: A dot is missing. “…17.86 + 4.38 nM (Fig3-C) This result was…”

-Line 339: Please rephrase the sentence “1 μg Vip1Ad (Vip2Ag)...” as it is almost impossible to understand.

-Section 4.8: Please, rewrite the first sentence.

Author Response

Response to Reviewer 3 Comments

Point 1:- Lines 75-76: Indicate that the yield obtained of each protein was high. This info “…and they are secretory proteins, which are easy to obtain” is part of the discussion not results. I totally agree on that it is important that the yield obtain was high but I think it would be great if you can put it in the discussion section and compare with other expression yields reported by other authors.

Response 1: Thanks for your suggestion. We moved this part about high expression to Discussion compared with the expression of, a common used strain, HD73 with red words.

Point 2:- Lines 90-91: “However, the corrected mortality of …” what it is the meaning of corrected mortality? Please rewrite the sentence to clarify it.

Response 2: Thanks for your suggestion. The modification has been made at Line 101-103 in red color.

Point 3:-Line 181: Please rewrite the sentence “... trans membraned to PVDF membranes…” Do you mean and transfer to PVDF membranes? In general, this section could be improved.

Response 3: Thanks for your suggestion. The modification has been made at Line 182-188 in red color.

Point 4:- Line 230: A dot is missing. “…17.86 + 4.38 nM (Fig3-C) This result was…”

Response 4: Thanks for your suggestion. The modification has been made at Line 241 in red color.

Point 5:-Line 339: Please rephrase the sentence “1 μg Vip1Ad (Vip2Ag)...” as it is almost impossible to understand.

Response 5: Thanks for your suggestion. The modification has been made at Line 349-352 in red color.

Point 6:-Section 4.8: Please, rewrite the first sentence.

Response 6: Thanks for your suggestion. The modification has been made at Line 369-373 in red color.

This manuscript is a resubmission of an earlier submission. The following is a list of the peer review reports and author responses from that submission.

Round 1

Reviewer 1 Report

This work is particularly interesting but varies widely weaknesses in the Chapter” Materials and Methods”. I believe that the whole chapter should be re-written from the beginning and become more understandable to the reader. Really it is very difficult to tell anybody how the experiments were made and the writers how implementation of experimental design. The writers must write a statistic chapter.

What do you mean by this statement free of pesticides field?

How did you know?

What Bt strains, are toxic to H. parallela larvae?

Author Response

Response to Reviewer 1 Comments

Point 1: This work is particularly interesting but varies widely weaknesses in the Chapter” Materials and Methods”. I believe that the whole chapter should be re-written from the beginning and become more understandable to the reader. Really it is very difficult to tell anybody how the experiments were made and the writers how implementation of experimental design. The writers must write a statistic chapter.

Response 1: Thank you for your comments about the Chapter” Material and Methods”. I re-wrote the chapter “Materials and Methods”. More details of the experiment and statistical analysis were added at paragraph 4.4. All revision was shown in the manuscript.

Point 2: What do you mean by this statement free of pesticides field? How did you know?

Response 2: Thank you very much for your suggestion. Yes, you are right, the pesticides were not used in that fields but it’s difficult to demonstrated, so we delete these words.

Point 3: What Bt strains, are toxic to H. parallela larvae?

Response 3: Thank you very much for your suggestion. The Bt185 (Changlong Shu, et. al. 2007, Curr. Microbiol. 55: 496-496) and HBF18 (Changlong Shu, et. al. 2009, Curr. Microbiol. 58: 389-392) and BIOT185 (Jingjing Liu, et. al. 2010, Appl. Microbiol. Biotechnol. 87:243-249) strains were toxic to the H. parallela larvae.

Reviewer 2 Report

              The authors assume that the mechanism of insecticidal activity of Vip1Ad and Vip2Ag is the "A + B model", and experiments are being conducted to confirm it. The manuscript has beautiful electron micrographs of BBMV and interesting data, however, because some experimental results are confusingly displayed, I recommend you modify the manuscript to make it easier to understand.

Although Vip1Ad changes from precursors to activated ones, and then forms multimers, the expression in the manuscript is unstable. It is clear that "Vip1Ad protoxin" is a precursor, but is it a precursor or an activated toxin if it is written "Vip1Ad"? And is Vip1A-T an activated toxin or trypsin-treated Vip1Ad? Trypsin-treated Vip1Ad seems to be a mixture of activated Vip1Ad, remaining trypsin and cleaved peptide fragments. Authors need to make these definitions more rigorous.

And is Vip2Ag protoxin (L108) the same as Vip2Ag or not? The authors should also mention in the introduction whether Vip2 needs the trypsin-treatment.

Fig. 5 is important data to confirm that Vip1Ad has receptor-like function, so it is necessary to display the state of binding in all combinations clearly. Specifically, it is necessary to prepare Vip1Ad, Vip1A-T, and Vip2Ag on a PVDF membrane, and perform ligand blotting with four types of ligands (Vip1Ad, total Vip1A-T, Vip1A-T monomer, and Vip2Ag), respectively. Of course, the indication of the oligomer band on the membrane should not be omitted. Similar experiments may be also required for the dot blot analysis.

Since there are many reports that Bt toxin forms multimers by SDS treatment, it is better to clarify whether multimers of activated Vip1Ad shown in Fig.4A were generated by BBMV or SDS treatment. For that purpose, purified activated Vip1Ad monomer should be included together in Fig 4A.

Moreover, I have some minor questions in this manuscript, and they are listed below. The authors should make them clear.

L14; There is no mention of "Vacuolization" in the results section.

L19; An abbreviation "Vip1A-T" is written without explanation.

L65; Although it is described that "the protein concentration was measured by BCA" in the method section, which one is correct?

L69; Which ion?

L76; Fig. 2A and 2C do not look like "step gradient elution".

Fig.2A and 2C have insufficient resolution. And they should be displayed so that the correspondence between the fraction of chromatography and each lane of SDS-PAGE can be understood.

Fig.2C shows that the culture solution contains no proteins or peptides other than Vip2Ag, is it correct?

L92; An abbreviation "HBF-18" is written without explanation.

L126; There is no Vip1Ad in Fig.5C.

Fig.5C; I do not know why the purification of Vip1A-T monomer takes place suddenly.

Fig.5D; Vip1Ad monomer has appeared suddenly without any explanation.

L136; I cannot find the SDS-PAGE result which shows "Vip1A-T monomers and oligomers". Possibly the second lane in Fig. 4A? However, it is unclear whether the bands above 180kDa in Fig.4A are Vip1Ad or Vip1A-T, may be aggregates between BBMV protein and Vip1Ad or Vip1A-T.

L137; What is "c(s)"?

L210; "ion affinity chromatography" is a less well-known method and needs to be referenced.

L224; Which concentration?

L266; The method of "blocking with PBS" is novel and should be referenced.

L281; Is "Institute of Biophysics, Chinese Academy of Sciences" the maker of "Proteome Lab XL-1"? If the authors borrowed the machine at the institute, it would be better to mention in the acknowledgment section.

That's all.

Author Response

Response to Reviewer 2 Comments

Point 1: Although Vip1Ad changes from precursors to activated ones, and then forms multimers, the expression in the manuscript is unstable. It is clear that "Vip1Ad protoxin" is a precursor, but is it a precursor or an activated toxin if it is written "Vip1Ad"? And is Vip1A-T an activated toxin or trypsin-treated Vip1Ad? Trypsin-treated Vip1Ad seems to be a mixture of activated Vip1Ad, remaining trypsin and cleaved peptide fragments. Authors need to make these definitions more rigorous.

Response 1: Thank you very much for your suggestion. Throughout the manuscript, Vip1Ad protein refers to the protoxin. Vip1A-T is trypsin treated Vip1Ad, consisting of activated Vip1Ad, remaining trypsin and cleaved peptide fragments.

Point 2: And is Vip2Ag protoxin (L108) the same as Vip2Ag or not? The authors should also mention in the introduction whether Vip2 needs the trypsin-treatment.

Response 2: hank you very much for your suggestion. Vip2Ag protoxin same as Vip2Ag. To avoid misunderstanding, we changed the “Vip2Ag protoxin” to “Vip2Ag” in the manuscript. In previous studies, trypsin-treatment of Vip2 was not reported. In present research, we found that Vip2Ag cannot be activitied by trypsin and the date was added in Figure S2.

Point 3: Fig. 5 is important data to confirm that Vip1Ad has receptor-like function, so it is necessary to display the state of binding in all combinations clearly. Specifically, it is necessary to prepare Vip1Ad, Vip1A-T, and Vip2Ag on a PVDF membrane, and perform ligand blotting with four types of ligands (Vip1Ad, total Vip1A-T, Vip1A-T monomer, and Vip2Ag), respectively. Of course, the indication of the oligomer band on the membrane should not be omitted. Similar experiments may be also required for the dot blot analysis.

Response 3: Thank you for comments about binding assay. The result of Vip1Ad (or Vip2Ag) on PVDF membrane incubated with Vip2Ag (or Vip1Ad) was shown in figure S1. Vip2Ag on the PVDF membrane were incubated by Vip1A-T (consists of monomer and oligomers) showing an obvious binding interaction (Figure 5-A). However, Vip1A-T protein on the membrane were incubated by Vip2Ag with no binding. Vip2Ag on the PVDF membrane were incubated by Vip1Ad monomer with no binding (figure 5-D).

Point 4: Since there are many reports that Bt toxin forms multimers by SDS treatment, it is better to clarify whether multimers of activated Vip1Ad shown in Fig.4A were generated by BBMV or SDS treatment. For that purpose, purified activated Vip1Ad monomer should be included together in Fig 4A

Response 4: Thank you very much for your suggestion. As you mentioned, many researchers reported that Bt toxin forms multimers by SDS treatment. For this work, Vip1Ad protoxin was incubated with BBMV and the signal of Vip1Ad monomer and oligomers were detected by antibody, so we think the multimers you mentioned were formed with BBMV.

Point 5: L14; There is no mention of "Vacuolization" in the results section.

Response 5: Thank you very much for your suggestion and we agree to this review. We have added vacuolization into paragraph 2.4. The modified manuscripts were marked in yellow.

Point 6: L19; An abbreviation "Vip1A-T" is written without explanation.

Response 6: Thank you very much for your suggestion and we agree to this review. We have explained Vip1A-T in Abstract. The modified manuscripts were marked in yellow.

Point 7: L65; Although it is described that the protein concentration was measured by BCA" in the method section, which one is correct?

Response 7: Thank you very much for your suggestion. the purpose of BCA method and optical densitometric method are different. The protein concentration measured by BCA was used to measure total concentration of purified proteins. While in paragraph 2.1, the concentration of Vip proteins extracts was determined by optical densitometry method, which calculate the concentration of specific band, the method has been added in the part of “method and materials 4.1”.

Point 8: L69; Which ion?

Response 8: Thank you very much for your suggestion. Vip1Ad and Vip2Ag proteins were eluted by different ions (Cl- for Vip1Ad and Na+ for Vip2Ad) and red line indicate conductivity. So,we re-written the “as the ion increased” to “as the conductivity increased”

Point 9: L76; Fig. 2A and 2C do not look like"step gradient elution".

Response 9: Sorry for my mistake, we used "linear gradient elution" not "step gradient elution". We revise the "step gradient elution" to "linear gradient elution".

Point 10: Fig.2A and 2C have insufficient resolution. And they should be displayed so that the correspondence between the fraction of chromatography and each lane of SDS-PAGE can be understood

Response 10: Thank you very much for your suggestion and we agree to this review. The figure 2 has been modified.

Point 11: Fig.2C shows that the culture solution contains no proteins or peptides other than Vip2Ag, is it correct?

Response 11: Thank you very much for your question. In Vip2Ag fermentation broth have other proteins or peptides, but their concentrations are low. In figure 1 and figure 2D, we can find shallower bands than Vip2Ag.

Point 12: L92; An abbreviation "HBF-18" is written without explanation.

Response 12: Thank you very much for your suggestion and we agree to this review. HBF-18 is a name of Bt strain screened by Shu et al., I have explained the "HBF-18" in manuscript.

Point 13: L126; There is no Vip1Ad in Fig.5C.

Response 13: Thank you very much for your suggestion and we agree to this review. We have revised figure 5C. Vip1Ad and Vip1A-T have been showed in figure S2. Here, we used Vip1A-T as a control.

Point 14: Fig.5C; I do not know why the purification of Vip1A-T monomer takes place suddenly.

Response 14: Thank you very much for your question. In figure 5A and 5B have showed that Vip1A-T and Vip2Ag could bind each other. Vip1A-T consist of monomer and oligomers, but we do not know which one could bind to Vip2Ag. So, we purified the monomer and found that it bound to Vip2Ag. The result showed that Vip1Ad monomer does not bind to Vip2Ag. We speculate that Vip1Ad oligomers bind to Vip2Ag.

Point 15: Fig.5D; Vip1Ad monomer has appeared suddenly without any explanation.

Response 15: Thank you very much for your suggestion and we agree to this review. The Vip1Ad monomer has been explained in paragraph 2.7.

Point 16: L136; I cannot find the SDS-PAGE result which shows "Vip1A-T monomers and oligomers". Possibly the second lane in Fig. 4A? However, it is unclear whether the bands above 180kDa in Fig.4A are Vip1Ad or Vip1A-T, may be aggregates between BBMV protein and Vip1Ad or Vip1A-T.

Response 16: Thank you very much for your suggestion. SDS-PAGE result showed "Vip1A-T monomers and oligomers" was added in figure 5,

Point 17: L137; What is "c(s)"?

Response 17: Thank you very much for your question. C(s) is the abbreviation for "continuous size-distribution". We have change C(s) to "continuous size-distribution" in manuscript.

Point 18: L210; "ion affinity chromatography" is a less well-known method and needs to be referenced.

Response 18: Thank you very much for your suggestion. Actually, the ion affinity chromatography is ion exchange chromatography.

Point 19: L224; Which concentration?

Response 19: Thank you very much for your suggestion. We have explained the concentration minutely in paragraph 4.3.

Point 20: The method of "blocking with PBS" is novel and should be referenced.

Response 20: Thank you very much for your suggestion. I think you have some misunderstanding about the membrane is blocked by PBS containing 2% Tween-20 instead of PBS alone, while we added reference here.

Point 21: L281; Is "Institute of Biophysics, Chinese Academy of Sciences" the maker of "Proteome Lab XL-1"? If the authors borrowed the machine at the institute, it would be better to mention in the acknowledgment section.

Response 21: Thank you very much for your suggestion. We borrowed the machine at the institute, and We have added the thanks in the acknowledgment section.  

Reviewer 3 Report

The authors are trying to demonstrate the importance of the binding of Vip1 in allowing Vip2 to help cause toxicity against Holotrichia parallela.  This is a pretty straightforward manuscript.  English is pretty good but still could use some improvement.  Several comments:

Table S1:  Would prefer that it not be included as supplemental.  Although this table merely servers to show that this binary toxin is toxic to H. parallela and that it takes both proteins to be toxic, This reviewer would request additional metrics typically associated with LC50 data such as slope +/- SEM plus a statistical parameter.  On lines 87-88 the authors state the buffer alone did not result in any insecticidal activity.  So control mortality was Zero percent?  Seems strange and unusual.

This reviewer is confused regarding Figure 4B and Figure 5B. It appears that Vip1A can help bind Vip2 to membrane regardless of whether or not Vip1A has been trypsinized.  Is that correct?  Please explain. 

Author Response

Response to Reviewer 3 Comments

Point 1: Table S1: Would prefer that it not be included as supplemental. Although this table merely servers to show that this binary toxin is toxic to H. parallela and that it takes both proteins to be toxic, this reviewer would request additional metrics typically associated with LC50 data such as slope +/- SEM plus a statistical parameter. On lines 87-88 the authors state the buffer alone did not result in any insecticidal activity. So, control mortality was Zero percent?  Seems strange and unusual.

Response 1: Thanks for your suggestion. We have moved table 1 from the supplementary material to paragraph 2.3 of the manuscript. And there are corresponding changes in the manuscript.

The Vip1Ad and Vip2Ag of 50 μg/g soil were used to preliminary screening, and the corrected mortality are 36.67% and 26.67%. We believe that the LC50 of the two toxins were higher than this concentration, but we didn't set the concentration gradient statistics for the specific LC50 date. So, we do not write slope +/- SEM in this paper.

We describe here as no result in insecticidal activity represents is that the mortality rate shown in the negative control is 13.3%.

Point 2: This reviewer is confused regarding Figure 4B and Figure 5B. It appears that Vip1A can help bind Vip2 to membrane regardless of whether or not Vip1A has been trypsinized.  Is that correct? Please explain.

Response 2: Thank you for your question. Vip1Ad protoxin does not bind to Vip2Ag (Figure S1), but Vip1A-T can bind to Vip2Ag (Figure 5-B). In figure 4-A, Vip1Ad protoxin was digested during incubation with BBMVs. So, we think that activated Vip1Ad help Vip2Ag to bind to membrane.

Reviewer 4 Report

In this manuscript, the authors investigated the insecticidal mechanisms of Vip1 and Vip2 toxins produced by B. thuringiensis. The authors expressed and purified the proteins from the fermentation medium and showed that the mixture of two proteins, but not individual proteins, is highly active against H. parallela larvae. They also showed that Vip1Ad binds to BBM vesicles. Interestingly, Vip2Ag alone bound very weakly to BBM, however, its binding was strongly increased in the presence of Vip1Ad toxin. Together, these results indicate that Vip1Ad toxin might function as a receptor for Vip2Ag toxin and that Vip1Ad and Vip2Ag operate as a binary toxin. This is a very well designed study that clarifies the action of Vip1Ad and Vip2Ag toxins and therefore fits the journal very well. I have some suggestions on how to improve the manuscript.

The authors are advised to provide some background for each of the experiments because in current form it is insufficiently explained. For example, paragraph 2.1. please describe what exactly was analyzed by the SDS-PAGE, how these proteins were expressed, what and how much was loaded on the gel. Also, if I understand correctly, the authors loaded fermentation broth on a gel, and to me, it is very surprising that they observe only single bands of their proteins. Fermentation broth contains lots of other proteins, so why we do not see these proteins on a gel? Please clarify.

Paragraph 2.2. Again, please give more details on the experiment. How the proteins were expressed, what and how much was loaded on the column.

Paragraph 2.4. and figure 3. Please describe how you define damage. How do we suppose to know that epithelium is damaged?

Paragraph 2.5. Please, describe why APN enzymatic activity was measured and why this is an indication of BBMV applicability for the experiments.

Paragraph 4.1. Describe what exactly recombinant strains means. How these genes were cloned and which plasmids were used, and what expression system was used.

Line 92. Define what HBF-18 is.

Lines 155, 182. Agricultural not agriculture.

Line 160. High not highly.

Line 2. functions.

Author Response

Response to Reviewer 4 Comments

Point 1: The authors are advised to provide some background for each of the experiments because in current form it is insufficiently explained. For example, paragraph 2.1. please describe what exactly was analyzed by the SDS-PAGE, how these proteins were expressed, what and how much was loaded on the gel. Also, if I understand correctly, the authors loaded fermentation broth on a gel, and to me, it is very surprising that they observe only single bands of their proteins. Fermentation broth contains lots of other proteins, so why we do not see these proteins on a gel? Please clarify.

Response 1: (1). Thanks for your suggestion. I have added many details into paragraph 2.1. The modified manuscripts were marked in yellow. (2). In fermentation broth there are other proteins or peptides, but their concentrations are low. In figure 1 and figure 2-D, we can find weak bands besides Vip2Ag.

Point 2: Paragraph 2.2. Again, please give more details on the experiment. How the proteins were expressed, what and how much was loaded on the column.

Response 2: Thanks for your suggestion. I have added many details into paragraph 2.2. The modified manuscripts were marked in yellow.

Point 3: Paragraph 2.4. and figure 3. Please describe how you define damage. How do we suppose to know that epithelium is damaged?

Response 3: Thanks for your question. Destruction of the midgut manifested as vacuolization of the cytoplasm and abscission of microvilli. In figure 3, we can find vacuolization of the cytoplasm and slight destruction of BBM were detected

Point 4: Please, describe why APN enzymatic activity was measured and why this is an indication of BBMV applicability for the experiments

Response 4: Thank you very much for your suggestion. Aminopeptidase N (APN) activity was monitored because APN is a very common enzyme located in BBMV and because APN is a putative receptor for many Bt toxins. I have added many details into paragraph 2.5. The modified manuscripts were marked in yellow.

Point 5: Paragraph 4.1. Describe what exactly recombinant strains means. How these genes were cloned and which plasmids were used, and what expression system was used.

Response 5: Thank you very much for your suggestion. I have added the information of recombinant strains in paragraph 4.1. The modified manuscripts were marked in yellow.

Point 6: Line 92. Define what HBF-18 is.

Response 6: Thank you very much for your suggestion. HBF-18 was conserved in China General Microbiological Culture Collection Center (CGMCC), and the bacteria preservation number is 2070.We have explained the "HBF-18" in manuscript.

Point 7: Lines 155, 182. Agricultural not agriculture. Line 160. High not highly. Line 2. Functions

Response 7: Sorry for my mistakes. We carefully reviewed all words in the text and figures. They were written in the right form.

Round 2

Reviewer 1 Report

Accept in present form

Reviewer 2 Report

This manuscript is not worthy of peer review because the authors have not responded sincerely to my last points.